# Effects of a School-Based Physical Activity Intervention for Obesity, Health-Related Physical Fitness, and Blood Pressure in Children with Intellectual Disability: A Randomized Controlled Trial

**DOI:** 10.3390/ijerph191912015

**Published:** 2022-09-22

**Authors:** Aiwei Wang, Danran Bu, Siyue Yu, Yan Sun, Jingjing Wang, Tinky Chin Ting Lee, Julien S. Baker, Yang Gao

**Affiliations:** 1College of Physical Education, Yangzhou University, Yangzhou 225127, China; 2Department of Sport, Physical Education and Health, Hong Kong Baptist University, Hong Kong, China; 3Department of Sport, Hubei University, Wuhan 430062, China; 4Hubei Institute of Sport Science, Wuhan 432025, China; 5JC School of Public Health and Primary Care, The Chinese University of Hong Kong, Hong Kong, China; 6Mass Sports Research Center, China Institute of Sport Science, Beijing 100061, China; 7Centre for Health and Exercise Science Research, Hong Kong Baptist University, Hong Kong, China

**Keywords:** children, intellectual disability, intervention, randomized controlled trial (RCT), physical activity, exercise, overweight, obesity, health-related fitness, blood pressure, China

## Abstract

Children with intellectual disability (ID) are more vulnerable to health conditions than their typically developing peers. Evidence of effective interventions is scarce. A randomized controlled trial was conducted in 30 overweight and obese children with intellectual disability (ID) in China to evaluate a 12-week school-based physical activity intervention for obesity, health-related physical fitness (HRPF), and blood pressure. The intervention consisted of 24 physical activity (PA) sessions (2 sessions/week, 60 min/session), with exercise intensity progressively increasing from a moderate level to a vigorous level. All participants were followed up for 12 weeks after the intervention period to evaluate sustained effects. Outcomes were repeatedly measured at baseline, after the intervention, and after follow-up. The intervention was effective in reducing some obesity-related outcomes (including weight and body mass index) and improving some HRPF-related outcomes (including the 6 min walk test and the 30 s sit-to-stand test), with the significant effects being sustained after the 12-week follow-up. No effect was observed on blood pressure. The findings of this study contribute to the development and implementation of PA interventions to reduce obesity and improve HRPF in children with ID.

## 1. Introduction

Childhood obesity accompanied with poor health-related physical fitness (HRPF) is an alarming public health issue and has attracted much attention in recent years [1]. It has been well documented that children with obesity and poor HRPF are more likely to suffer from a series of health conditions, such as cardiovascular diseases, metabolic syndrome, type 2 diabetes, and hypertension [2,3,4]. Additionally, there is clear evidence that childhood obesity may lead to adult obesity [5]; poor HRPF in childhood may continue into adulthood; and the health conditions associated with obesity and poor HRPF in childhood may persist into adulthood as well [5,6].

Children with intellectual disabilities (ID), characterized by significant limitations in intellectual and adaptive functioning and accounting for 1.83% of the entire global pediatric population, are more vulnerable to obesity and lower levels of HRPF than their counterparts without ID [7,8]. Previous studies in different countries (including ours) consistently revealed prevalence rates of overweight and obesity of 30~33% in children with ID, about 1.54~1.80 times of those observed in their typically developing peers [8,9,10]. Lower levels of HRPF were also observed in children with ID compared with those without ID. A recent cross-sectional study among 128 children with ID identified that 71% to 91% of them scored below the age- and gender-specific reference values of HRPF, including cardiopulmonary fitness and muscular fitness [7]. Though evidence was limited, hypertension might also hit an alarming level, as shown in our previous cross-sectional study, where 31.4% of children with ID were hypertensive [11].

There are numerous factors associated with these conditions in general pediatric populations, including genetics, metabolic factors, medications, physical activity (PA), diet, etc. [12]. Except for these common factors, the higher prevalence of obesity, hypertension, and lower HRPF levels in children with ID may also be attributable to their limited mental ability to choose and adopt healthy behaviors [13].

Though children with ID have higher risks for obesity, lower levels of HRPF, and hypertension, intervention studies for this special population are scarce [14]. Evidence of effective interventions obtained in typically developing children may not be applicable to them, given the additional risks they are suffering due to limited intellectual functioning [13]. Thus, there is an urgent need to develop effective interventions for this special pediatric population. Previous evidence for children with intellectual disability suggested that school-based PA was the predominant intervention approach adopted and might contribute to reducing obesity and improving fitness [15]. Therefore, this study aimed to evaluate the effectiveness of a 12-week school-based PA intervention to reduce obesity- and hypertension-related indicators, and improve HRPF levels in Chinese children with ID.

## 2. Materials and Methods

### 2.1. Study Design and Participants

This study was a randomized controlled trial (RCT) with two groups, including an intervention group (IG) and a wait-list control group (CG), which was developed based on the findings of our systematic review and meta-analysis aiming to identify effective lifestyle interventions in children with ID [15], along with exercise guidelines recommended by the American College of Sports Medicine [2]. It consisted of a 12-week school-based PA intervention and a 12-week follow-up. Ethical approval was obtained from the Research Ethics Committee in Hong Kong Baptist University (Ref. No.: SPE17482062). The study was registered at the ClinicalTrials.gov (Ref. No.: NCT04554355) and the research protocol was published elsewhere [16]. The Consolidated Standard of Reporting Trials (CONSORT) 2010 guideline was followed for reporting the study [17] and the complete CONSORT 2010 checklist is shown in Appendix A.

The study was implemented in four special schools in China between June 2020 and April 2021. A sample size of 30 participants (15 in each group) was estimated using the G*Power (Version 3.1.9.4, created by Axel Buchner, Edgar Erdfelder, Franz Faul, and Albert-Georg Lang, Germany), with an alpha of 0.05, a power of 80%, an effect size (Cohen’s f) of 0.27 for body mass index (BMI, a primary outcome) [18], and a dropout rate of 20% [19]. Inclusion criteria of participants were (1) with ID; (2) aged 12–18 years old; (3) overweight or obese; and (4) at least one family member able to respond to study questionnaires. Exclusion criteria included (1) with any physical disability; (2) with a medical predisposition towards obesity (such as genetic syndrome) that could interfere with the results of the study; (3) with contraindications for PA (e.g., severe heart disease); and (4) participated in other obesity or fitness-related programmes in the past six months. PE teachers in the study schools screened for eligibility and recruited the participants, with written parental informed consent obtained in advance. All participants were randomly assigned (1:1 ratio) in either the IG or the CG using block randomization [20]. Both the participants and the assessors were blinded to the allocation results.

### 2.2. Intervention

There was a total of 24 sessions in the PA intervention, delivered at a frequency of twice a week with each session of 60 min. All sessions were divided into three levels and each level lasted for four weeks. Exercise intensity increased progressively from 40% heart rate reserve (HRR, moderate intensity) at Level 1 to 70% HRR (vigorous intensity) at Level 3 [2]. Each session consisted of a 10 min warm-up, a 45 min main exercise (including two 15 min aerobic games, followed by 15 min resistance training), and a 5 min cool-down. Twelve aerobic games were selected and then modified from the Jockey Club Keep Fit Formular for Children Programme with the following considerations: (1) achieving the intensity requirements; (2) being safe, simple, and interesting; and (3) being feasible to carry out at school. Detailed contents of the intervention are reported in the research protocol [16].

Target exercise heart rates (target HRs) of the participants were estimated before each training level using the equations listed in Table 1 [21]. Their real-time exercise heart rates (real-time HRs) were monitored by tutors twice in each training session by counting beats on their wrists in ten seconds and then multiplying by six. The real-time HRs were used to compare with the target HRs to determine if the participants met the requirements or not. If necessary, appropriate modifications (such as encouraging them to run faster/jump higher) were implemented to ensure their compliance with target HRs.

The PA programmes were conducted in the participants’ schools. A fixed sports area in each participating school was provided. The training session was arranged in fixed non-class timeslots during the school activity and implemented by well-trained tutors who were educated in sports sciences and had experience in teaching students with special needs. A tutor-to-participant ratio of 1:3 was applied to ensure that each participant received sufficient supervision. In addition, the PE teacher from each participating school was on site to assist with the delivery of the intervention. All participants (in both groups) were not permitted to join other PA or obesity-related programmes.

### 2.3. Wait-List Control

The participants in the wait-list control group received no intervention. They were asked to remain their PA and eating habits and not to join any other programs aiming to reduce obesity and/or improve HRPF. After the completion of the study (i.e., from April to June 2022), all participants in the control group received the same intervention. In addition, the programme manual and other materials were shared with the four participating special schools.

### 2.4. Measurements

Obesity-related and HRPF-related outcomes were measured as primary outcomes, whilst systolic blood pressure and diastolic blood pressure were involved as secondary outcomes. Children’s background information, health-related behaviors (including PA, sedentary behavior, sleep, and eating) and pubertal stage were collected via questionnaire and used as control variables. All measurements were repeated three times at pre-intervention (T1), post-intervention (T2), and after the follow-up (T3). A detailed description was reported elsewhere [16].

In brief, obesity-related outcomes consisted of: (1) weight (kg), BMI (kg/m^2^), and body fat percentage (%), measured using a TANITA digital scale (TBF-410) and a height gauge (SECA 0123); (2) waist circumference (cm) and waist-to-height ratio (%), measured using a flexible metre ribbon. HRPF-related outcomes were assessed with the 6 min walk test (6MWT, m, for cardiopulmonary fitness), 30 s sit-to-stand test (repetitions, for muscular strength and endurance of lower limbs), 1 min sit-up test (repetitions, for abdominal muscular strength and endurance), handgrip strength test (kg, for muscular strength and endurance of hands and forearms), and sit-and-reach test (cm, for flexibility). Systolic blood pressure and diastolic blood pressure (mmHg) were measured using an Omron blood pressure monitor (HBP-9020) following a standard protocol. In terms of the control variables, PA (daily minutes) and sedentary behavior (daily hours) were measured using a modified Chinese version of the Global Physical Activity Questionnaire (GPAQ), sleep duration (daily hours) was asked in a typical week, eating habits were assessed with a dietary questionnaire developed by the Central Health Education Unit of Hong Kong for school-aged children, and the pubertal stage was estimated with an illustrated Tanner pubertal questionnaires [9,11,22,23,24]. Possible adverse effects induced by the PA interventions, such as injuries and eating disorders, were monitored and recorded during the 24-week study period.

### 2.5. Process Evaluation

The participants’ retention rate, adherence rate, and compliance with the intervention were assessed. In addition, parents and schoolteachers were asked about their satisfaction with the intervention, perceived effectiveness and usefulness of the intervention, future participation intention, and intention to recommend the intervention to others in a 5-point Likert questionnaire [25].

### 2.6. Data Analysis

SPSS 23.0 (IBM Corp, Armonk, NY, USA) was used for data analysis. Independent variables consisted of group (the between-subjects: IG and CG) and time (the within-subjects IV: T1, T2, and T3). Dependent variables included primary outcomes (obesity-related outcomes and fitness-related outcomes) and secondary outcomes (blood pressure). Control variables consisted of baseline outcomes and lifestyle confounders. Mean with standard deviation and number with percentage were used to describe the distribution of continuous and categorical variables respectively. Independent t-test and Fisher’s exact test were applied to test between-group differences in dependent variables and control variables at baseline respectively.

Within-group differences for each group across the three time-points (T1, T2, and T3) were examined respectively using one-way analysis of variance (ANOVA) for repeated measures. Whenever there was a violation in the sphericity assumption, the Greenhouse–Geisser correction was used. Post-hoc analysis using Bonferroni correction was carried out to explore the differences between T1 vs. T2, T1 vs. T3, and T2 vs. T3. Between-group differences in changes from T1 to T2 and T3 were examined separately using analysis of covariance (ANCOVA) [26]. Covariates in the analyses included (1) baseline outcome values; (2) lifestyle confounders, if any significant between-group differences were found at three-time points. According to guidelines of **Cohen (1988)**, partial η^2^ = 0.01, 0.06, and 0.14 represent small, medium, and large effects respectively [27].

## 3. Results

A total of 42 children were assessed for eligibility and 12 of them were excluded. The remaining 30 participants were equally and randomly assigned to two groups and all of them completed the study (Figure 1).

Table 2 presents the characteristics of the participants at baseline. The mean age of the participants was 14.17 years (SD: 0.45 years) and most of them were male (n = 22, 73.33%), overweight (n = 23, 76.70%), and with mild and moderate ID levels (n = 24, 80.00%). Their BMI values at baseline ranged from 21.70 to 31.80 kg/m^2^ (Mean = 26.30 kg/m^2^, SD = 2.80 kg/m^2^). Few participants were living with autism (n = 4, 13.33%) and Down syndrome (n = 4, 13.33%). No other comorbidities existed among the participants. No significant difference between the two groups was observed at baseline.

### 3.1. Lifestyle Confounders of the Participants at Three-Time Points

The distribution of the lifestyle confounders at baseline, post-intervention, and post follow-up are listed in Appendix B. At baseline, most participants were physically inactive (n = 29, 96.67%), had sufficient sleep duration (n = 26, 86.67%), consumed insufficient fruit (n = 23, 76.67%), had less high-salt foods (n = 22, 73.33%), consumed more snacks (n = 25, 83.33%), and had breakfast everyday (n = 29, 96.67%). No significant between-group difference was found in any lifestyle confounder at any time point, except for puberty at T3 (*p* = 0.034, Fisher’s exact test). Thus, puberty was used as a covariate in the ANCOVA to examine between-group differences in the outcome variables.

### 3.2. Intervention Effects on Obesity-Related Outcomes

Table 3 presents the within-group changes in obesity-related outcomes from T1 to T2, T1 to T3, and T2 to T3 using one-way repeated measure ANOVA. Significant reductions in weight and BMI were found from T1 to T2 and from T1 to T3 in the IG (mean differences in weight: 1.00–1.10 kg, mean differences in BMI: 0.43–0.44 kg/m^2^), while their changes in the CG was insignificant (Table 3). In addition, significant increases in waist circumference, waist-to-height ratio, and body fat percentage were observed in the IG during the follow-up period (from T2 to T3), while comparisons in other time periods and those in the CG did not reach significance.

Table 4 shows the between-group changes in obesity-related outcomes at T2 and T3 (by ANCOVA). Similar to the results of the within-group analysis, significant reductions in weight (−1.15 kg and −1.10 kg) and BMI (−0.48 kg/m^2^ and −0.47 kg/m^2^) were found from T1 to T2 and T3 respectively. The between-group differences in other obesity-related changes were insignificant. Figure 2 illustrates the overall trends from T1 to T3 in obesity-related outcomes for both groups.

### 3.3. Intervention Effects on HRPF-Related Outcomes

Table 5 presents the within-group differences in HRPF-related outcomes at each time point, while Table 6 shows the between-group differences in HRPF-related outcomes at T2 and T3. In the IG, intervention induced improvements reached significance in 6MWT, 30 s sit-to-stand, 1 min sit-ups, and handgrip strength, which were retained after the follow-up, while changes in the sit-and-reach were insignificant. Results of the ANCOVA for between-group differences revealed similar results in 6MWT and 30 s sit-to-stand (i.e., significant and sustainable improvements), while the improvements in 1 min sit-ups became insignificant (at both T2 and T3) and significant improvement in handgrip strength (at T2) became unsustainable (at T3). Figure 3 presents mean values for all HRPF-related variables of both groups across the three time points.

### 3.4. Intervention Effects on Blood Pressure

Table 7 and Table 8 list the within-group differences and between-group differences in blood pressure respectively. No result achieved significance. The overall trends in the blood pressure for both groups are illustrated in Figure 4.

### 3.5. Adverse Effects

No participants reported adverse effects of the intervention during the entire study period.

### 3.6. Process Evaluation

The study had a retention rate of 100.00% and an overall attendance rate of 96.94%. Appendix C elaborates the attendance rate in each training session. Regarding the effectiveness, usefulness, and satisfaction of the programme, most parents/guardians (>80%) provided positive feedback (including “Agree” and “Strongly agree”), as shown in Appendix D. In addition, all four teachers (with one from each school) rated all items about effectiveness, arrangement, and staffing of the intervention positively and were willing to participate in the future and recommend it to other special schools (Appendix E).

## 4. Discussion

This study developed and evaluated a school-based PA intervention for reducing obesity-related indicators, improving HRPF levels, and reducing blood pressure in 30 overweight and obese children with ID. The intervention consisted of 24 60-min sessions delivered twice per week for 12 weeks. Both aerobic games and resistance training were offered in each session, with exercise intensity progressively increasing from moderate (40%HRR) to vigorous (70%HRR). The intervention was effective in reducing some obesity-related outcomes (including weight and BMI) and improving some HRPF-related outcomes (including the 6MWT and the 30 s sit-to-stand test), with the significant effects being sustained for 12 weeks after completion of the intervention, as shown in the ANCOVA results. Handgrip strength was significantly improved right after the intervention. However, the effect reduced and became insignificant after the follow-up. The intervention was ineffective in changing blood pressure. During the study period, no participant reported any adverse effects. In addition, the retention and attendance rates were high. Feedback regarding effectiveness and arrangement from parents and teachers was positive.

### 4.1. Obesity-Related Outcomes

The participants in the IG reported a significant reduction in weight and BMI at T2 compared with the participants in the CG. This finding is consistent with previous research, specifically with three studies evaluating the effects of similar short-term PA programmes on reducing weight and BMI in children with ID [18,28,29]. However, different from the three studies that used a training frequency of three or five sessions per week, we performed two sessions per week. This suggests that a training frequency as low as twice per week may be sufficient to reduce obesity. Previous research has suggested that participant retention and attendance may reduce along with frequency increase and high frequency is a risk factor for injuries [29]. Our results of the 100% retention rate, 96.9% attendance rate, and zero injuries lent support to this evidence. In addition, two [18,29] of the three studies followed a routine with higher vigorous exercise intensity (60–75%HRR) compared with this study (from moderate (40%HRR) to vigorous (70%HRR)). However, such a vigorous exercise routine is not recommended for children with ID, as this population typically has poor motor skills, which may constrain their exercise compliance, especially for inactive children and at the early stage of training [30]. Thus, studies with exercise intensity progressively improving, such as our study, may be safer and provide more health benefits for children with ID.

Apart from weight and BMI (the most widely used measures for obesity), other obesity-related outcomes (including waist circumference, waist-to-height ratio, and body fat percentage) were also included in this study to estimate the study effects on body fat distribution [31]. Although consistent reductions were observed in these central adiposity indicators, none of them reached statistical significance. This finding is also consistent with previous studies, which have attributed this non-significance to insufficient intervention duration [32]. Furthermore, the sample size of this study (N = 30) was estimated using BMI as the outcome, which may be insufficient to obtain significant reductions in these variables if their effect sizes are smaller than BMI. Thus, long-term interventions and those with a large sample size are warranted to examine the effects on these outcomes.

### 4.2. HRPF-Related Outcomes

In this study, cardiopulmonary fitness was examined by the 6MWT. Our findings suggest that the 12-week PA intervention can improve the 6MWT distance (partial η^2^ = 0.149), which is consistent with our previous findings from a systematic review and meta-analysis for children with ID [15]. In that systematic review and meta-analysis, seven out of eight PA interventions employing the 6MWT reported effective results on cardiopulmonary fitness [18,29,33,34,35,36,37] (partial η^2^ = 0.010–0.600). Among those reporting effective (including this study), the interventions with vigorous intensity (compared with moderate intensity), those following the progression principle (compared with constant intensity), and those combined aerobic and resistance training (compared with aerobic training only) tended to generate larger effect sizes (partial η^2^ > 0.14) [18,29,33,35].

Several tests were employed in this study to examine the intervention effects on muscular strength and endurance, of which significant improvements were achieved in the 30 s sit-to-stand test for lower limb strength and endurance and the handgrip strength test for upper limb strength and endurance, but not in the 1 min sit-ups test for abdominal muscular strength and endurance. To our best knowledge, there were three previous interventions for children with ID that employed the 30 s sit-to-stand test [18,29,38]. All of them revealed significant positive effects, with partial η^2^ ranging from 0.010 to 0.170 across the individual studies. Compared with those studies, we observed a larger effect (partial η^2^ = 0.244). The superior treatment effects in our study might be due to the progressive increments in the exercise intensity, which was not the case in those studies. Four interventions measured handgrip strength [18,29,35,37], while only one was reported effective (partial η^2^ = 0.0002) [18]. Our study achieved a large effect (partial η^2^ = 0.276). Considering the specificity principle of training, the significant improvement in handgrip strength in our study may be attributable to the involvement of exercises for the upper limbs (e.g., picking up bean bags and dribbling and throwing balls). We failed to observe a significant effect on abdominal muscular strength and endurance in the 1 min sit-up test, though significance was reached in changes in the IG without considering the control. The relatively small number of participants may play a role in the insignificant outcome. In addition, similar to this finding, three previous studies with the same 12-week intervention period as ours also reported it as ineffective [28,37,39]. Future interventions with a longer intervention duration than 12 weeks are therefore promising to obtain effective results in the 1 min sit-up test [40].

Flexibility, measured by the sit-and-reach test, was not significantly improved in this study. Findings from previous studies on this were mixed. In our previous systematic review and meta-analysis, only two out of six studies involving this test observed significant improvements [15]. The two studies differed in study designs and exercise protocols, and it is therefore impossible to recommend effective intervention elements for improving flexibility. However, it is possible that the studies with negative results (including our study) were not adequately focused on flexibility, which is something that future studies can rectify and explore.

### 4.3. Blood Pressure

Against our hypothesis, we did not observe any significant improvement in blood pressure. The effects of exercise on treating and preventing hypertension in populations without ID have been well established, with the recommended routine being moderately intense aerobic exercise for 30 min on most days supplemented by resistance training two to three days per week [41]. However, evidence in children with ID was scarce. We found only one previous study examining the intervention effect on blood pressure [33]. That study included two intervention groups, a sprint interval training group, and a continuous aerobic exercise group. Positive effects were achieved in the sprint interval training group only. As sprint interval training is an extremely effort-intensive [42], whether such a programme is practicable and sustainable for the population with ID remains under-researched.

### 4.4. Sustained Effects

In this study, the participants were followed up for 12 weeks to examine the sustained effects of the intervention. First, significant body weight and BMI reductions were maintained at the follow-up examination. These findings provide new insight into the sustained effects of PA intervention on obesity-related outcomes, as no previous research has examined this aspect. Such sustainability strengthens the intervention effects on reducing obesity in children with ID. Future studies are suggested to examine the sustained effects over a longer follow-up period. Secondly, the significantly improved cardiopulmonary fitness (indicated by distance covered in the 6MWT) and muscular strength and endurance of the lower limbs (indicated by complete repetitions in the 30 s sit-to-stand test) right after the intervention were also maintained after the follow-up. To date, two studies evaluated the sustained effects of PA interventions on cardiopulmonary fitness, one reporting positive results after a 24-week follow-up [43] and the other reporting no effect sustained after one year of follow-up [44]. No study evaluated the sustained effects on lower limb muscular strength and endurance. Based on the limited evidence, we suggest that the intervention induced improvements in HRPF measures would vanish over time. It is therefore of importance in future studies to consider how to maintain PA levels after the PA interventions, while those supported with school policy changes (e.g., providing more PE classes), those with parental engagement (e.g., doing exercise with parents), and those integrated with behavior change components (e.g., increasing exercise self-efficacy) may be promising. Given that the academic demands in special schools are not as high as in mainstream schools, PA interventions with school policy support are therefore superior.

### 4.5. Strengths and Limitations

The strengths of the study include (1) the intervention was developed based on findings from our systematic review and meta-analysis, aiming at identifying effective interventions for reducing obesity and improving HRPF in children with ID [15]; (2) the study achieved very high retention and attendance rates. The use of fun aerobic games may have improved the children’s motivation for participation, which also contributed to the high retention and attendance rates; (3) the intervention contents were evaluated by experts before implementation, which ensured rationality, suitability, and operability of the study; (4) puberty and lifestyle confounders were considered in this study, and their possible effects on the study outcomes were therefore ruled out; (5) the RCT design ensured the validity of the study results.

However, the study has the following limitations: (1) the sample size, estimated using BMI, was small. Thus, it may not be sufficient for other outcomes, and therefore limited the power to identify significant differences in changes in those outcomes; (2) the confounding variables (e.g., moving habits, eating habits) were collected via a questionnaire, which may have resulted in report bias and recall bias; (3) some medication may act as a risk factor of weight gain [45]. However, we did not collect this information, and therefore could not know its influences on obesity-related effects. (4) The study population was Chinese children with ID. As a result, our findings may not be generalized to non-Chinese populations.

## 5. Conclusions

The 12-week PA intervention was found effective in reducing the degree of obesity in children with ID, as weight and BMI in the IG decreased significantly compared with the CG. These effects were sustained 12 weeks after the intervention. In addition, the intervention effectively improved cardiopulmonary fitness, lower limb muscular strength and endurance, and hand and forearm strength, while only the effect on hand and forearm strength did not sustain at the follow-up evaluation. Moreover, the intervention was ineffective on blood pressure. Overall, this study provides practicable evidence for reducing obesity and improving HRPF in children with ID. Future PA interventions with large sample sizes, specifically designed for study objectives, and supported by school policy changes are suggested to elucidate for this special population.

## Figures and Tables

**Figure 1 ijerph-19-12015-f001:**
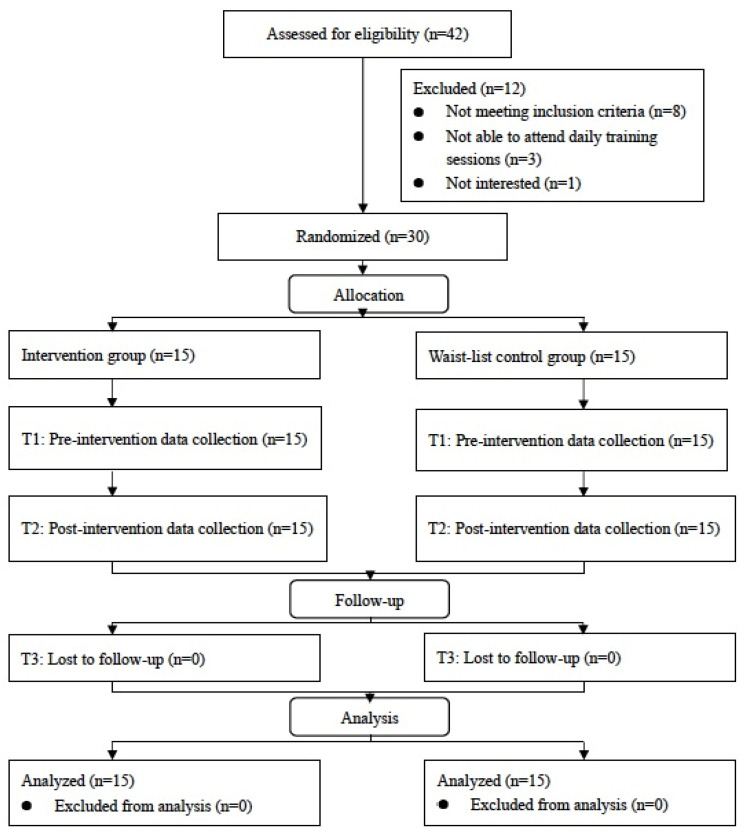
Participant flow diagram in CONSORT format.

**Figure 2 ijerph-19-12015-f002:**
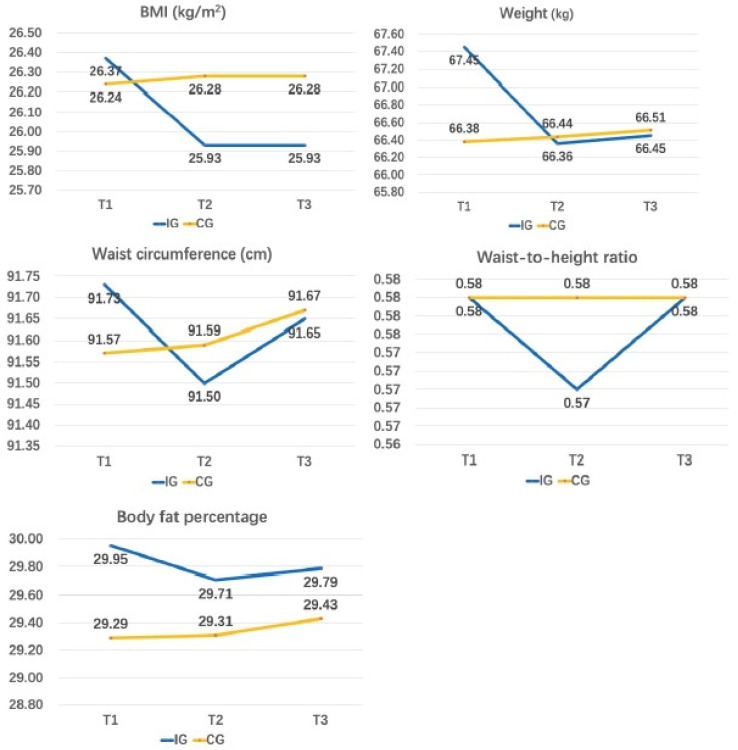
Changes in obesity-related outcomes across the three-time points.

**Figure 3 ijerph-19-12015-f003:**
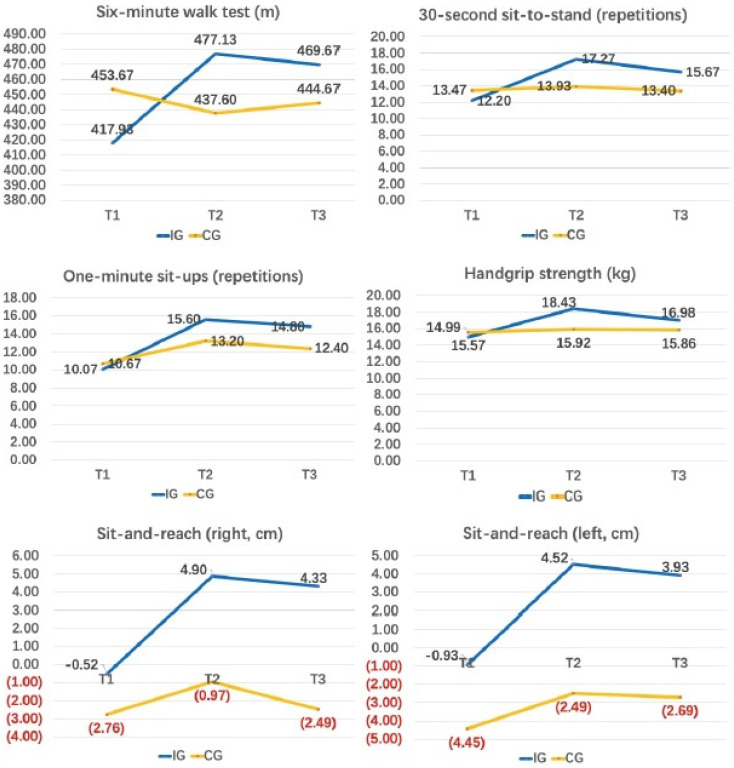
Changes in HRPF-related outcomes across the three-time points.

**Figure 4 ijerph-19-12015-f004:**
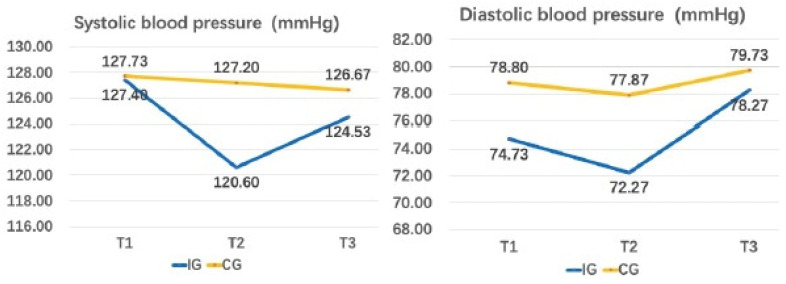
Changes in blood pressure across the three-time points.

**Table 1 ijerph-19-12015-t001:** Equations for calculating target exercise heart rate [21].

No.	Equation
(1)	Exercise HR ^1^ (beats/min) = Target% × HRR ^2^ + HR_rest_ ^3^
(2)	HRR (beats/min) = HR_max_ − HR_rest_
(3)	HR_max_ ^4^ (beats/min) = 210 − 0.56 × age (in years) − 15.5 (Down syndrome) *Down syndrome coded as 2; non-Down syndrome coded as 1*

*Note.* ^1^ HR, heart rate; ^2^ HRR, heart rate reserve; ^3^ HR_rest_, resting heart rate; ^4^ HR_max_, maximal heart rate.

**Table 2 ijerph-19-12015-t002:** Participant characteristics at baseline.

Variable	Total N = 30	IG ^1^ n = 15	CG ^2^ n = 15	*p* ^3^
**Age (range 12–18 years), Mean ± SD ^4^**	14.17 ± 0.45	14.60 ± 2.20	13.73 ± 1.44	0.212
**Gender, n (%)** - Male - Female	22 (73.3) 8 (26.7)	9 (60.0) 6 (40.0)	13 (86.7) 2 (13.3)	0.215
**BMI (range 21.70–31.80 kg/m^2^), Mean ± SD ^4^**	26.30 ± 2.80	26.37 ± 2.74	26.24 ± 2.95	0.745
**BMI category, n (%)** - Overweight - Obesity	23 (76.7) 7 (23.3)	12 (80.0) 3 (20.0)	11 (73.3) 4 (26.7)	1.000
**ID level, n (%)** - Mild - Moderate - Severe	14 (46.7) 10 (33.3) 6 (20.0)	7 (46.7) 6 (40.0) 2 (13.3)	7 (46.7) 4 (26.7) 4 (26.7)	0.580
**Comorbidity, n (%)**				
- Autism Spectrum Disorder ✓ Yes ✓ No	4 (13.3) 26 (86.7)	2 (13.3) 13 (86.7)	2 (13.3) 13 (86.7)	1.000
- Down Syndrome ✓ Yes ✓ No	4 (13.3) 26 (86.7)	3 (20.0) 12 (80.0)	1 (6.7) 14 (93.3)	0.598
**Special school, n (%)**				
- 1	7 (23.3)	4 (26.7)	3 (20.0)	0.923
- 2	8 (26.7)	4 (26.7)	4 (26.7)	
- 3	7 (23.3)	4 (26.7)	3 (20.0)	
- 4	8 (26.7)	3 (20.0)	5 (33.3)	

*Note.* ^1^ IG, intervention group; ^2^ CG, wait-list control group; ^3^ To detect between-group differences, independent sample *t*-test was performed for continuous variables and Fisher’s exact test was performed for categorical variables; ^4^ SD, standard deviation.

**Table 3 ijerph-19-12015-t003:** Within-group differences in obesity-related outcomes.

Outcome	Change from T1 ^1^ to T2 ^2^ (T2–T1)		Change from T1 to T3 ^3^ (T3–T1)		Change from T2 to T3 (T3–T2)	
	Mean Difference (95%CI) ^4^	*p* ^5^	Mean Difference (95%CI)	*p*	Mean Difference (95%CI)	*p*
**Weight (kg)** - IG ^6^ - CG ^7^	−1.10 (−1.77, −0.43) 0.06 (−0.22, 0.34)	**0.003** 0.652	−1.00 (−1.74, −0.26) 0.13 (−0.31, 0.56)	**0.003** 0.542	−0.10 (−0.20, 0.40) 0.07 (−0.42, 0.56)	0.480 0.774
**BMI ^8^ (kg/m^2^)** - IG - CG	−0.44 (−0.68, −0.20) 0.04 (−0.08, 0.16)	**0.002** 0.472	−0.43 (−0.66, −0.20) 0.04 (−0.13, 0.21)	**0.001** 0.619	0.01(−0.12, 0.13) 0.09 (−0.17, 0.18)	0.894 0.994
**Waist circumference (cm)** - IG - CG	−0.23 (−0.81, 0.34) 0.01 (−0.24, 0.26)	0.396 0.910	−0.08 (0.47, −0.63) 0.09 (−0.22, 0.41)	0.760 0.533	0.15 (0.23, 0.08) 0.08 (−0.19, 0.35)	**0.001** 0.533
**Waist-to-height ratio** - IG - CG	−0.001 (−0.005, 0.002) 0.001 **(−0.0014, 0.0016)**	0.449 0.888	−0.001 (−0.004, 0.003) 0.0004 **(−0.0019, 0.0026)**	0.709 0.738	0.001 (0.00003, 0.001) 0.0003 **(−0.0017, 0.0022)**	**0.042** 0.790
**Body fat percentage** - IG - CG	−0.24 (−0.70, 0.22) 0.02 (−0.37, 0.41)	0.282 0.914	−0.16 (−0.60, 0.28) 0.14 (−0.34, 0.62)	0.445 0.538	0.08 (−0.15, 0.31) 0.12 (−0.07, 0.31)	0.471 0.191

*Note.* ^1^ T1, pre-intervention; ^2^ T2, post-intervention; ^3^ T3, follow-up; ^4^ 95%CI, 95% confidence interval; ^5^ one-way repeated measure ANOVA was performed to detect within-group differences, and post-hoc analysis with Bofferoni correction was employed to test significance of the changes from T1 to T2, T1 to T3, and T2 to T3; ^6^ IG, intervention group; ^7^ CG, wait-list control group; ^8^ BMI, body mass index; bold text indicates significant results.

**Table 4 ijerph-19-12015-t004:** Between-group differences in obesity-related outcomes.

Outcome	IG ^1^ (n = 15) Mean ± SD ^3^	CG ^2^ (n = 15) Mean ± SD	Adjusted Difference ^4^ (95%CI) ^5^	*p*	Partial η^2^
**Weight (kg)** - T1 - T2 - T3	67.45 ± 11.12 66.35 ± 10.91 66.45 ± 11.11	66.38 ± 14.03 66.44 ± 13.88 66.51 ± 13.63	−1.15 (−1.83, −0.46) −1.10 (−1.79, −0.42)	**0.002** **0.003**	0.303 0.297
**BMI ^6^ (kg/m^2)^** - T1 - T2 - T3	26.37 ± 2.74 25.93 ± 2.73 25.94 ± 2.73	26.24 ± 2.95 26.28 ± 2.96 26.28 ± 2.79	−0.48 (−0.74, −0.22) −0.47 (−0.74, −0.21)	**0.001** **0.001**	0.344 0.348
**Waist circumference (cm)** - T1 - T2 - T3	91.73 ± 5.36 91.50 ± 5.02 91.87 ± 9.42	91.57 ± 5.79 91.59 ± 5.69 92.83 ± 8.16	−0.24 (−0.81, 0.33) −0.16 (−0.73, 0.40)	0.399 0.554	0.027 0.014
**Waist-to-height ratio** - T1 - T2 - T3	0.58 ± 0.04 0.57 ± 0.04 0.58 ± 0.04	0.58 ± 0.05 0.58 ± 0.05 0.58 ± 0.05	−0.01 (−0.05, 0.04) −0.02 (−0.03, 0.01)	0.761 0.567	0.003 0.013
**Body fat percentage** - T1 - T2 - T3	29.95 ± 8.47 29.71 ± 8.24 29.79 ± 8.43	29.29 ± 6.48 29.31 ± 6.56 29.43 ± 6.63	−0.25 (−0.83, 0.33) −0.30 (−0.94, 0.34)	0.387 0.347	0.028 0.034

*Note.* ^1^ IG, intervention group; ^2^ CG, wait-list control group; ^3^ SD, standard deviation; ^4^ Adjusted difference, the difference of the intervention group relative to the wait-list control group in ANCOVA, adjusted for covariates (baseline values were adjusted in the between-group difference at T2; baseline values and changes in puberty stage were adjusted in the between-group difference at T3); ^5^ 95%CI, 95% confidence interval; ^6^ BMI, body mass index; bold text indicates significant results.

**Table 5 ijerph-19-12015-t005:** Within-group differences in HRPF-related outcomes.

Outcome	Change from T1 ^1^ to T2 ^2^ (T2–T1)		Change from T1 to T3 ^3^ (T3–T1)		Change from T2 to T3 (T3–T2)	
	Mean Difference (95%CI) ^4^	*p* ^5^	Mean Difference (95%CI)	*p*	Mean Difference (95%CI)	*p*
**6MWT ^6^ (m)** - IG ^7^ - CG ^8^	59.20 (27.19, 91.22) −16.07 (−73.98, 41.85)	**0.001** 0.561	51.73 (20.39, 83.08) −9.00 (−47.79, 29.79)	**0.003** 0.626	−7.47 (−34.61, 19.68) 7.07 (−26.86, 41.10)	0.565 0.662
**30 s sit-to-stand (repetitions)** - IG - CG	5.07 (2.40, 7.74) 0.47 (−1.27, 2.20)	**0.001** 0.574	3.47 (1.63, 5.30) −0.07 (−1.10, 0.97)	**0.001** 0.483	−1.60 (−3.57, 0.37) −0.53 (−1.80, 0.74)	0.104 0.383
**1 min sit-ups (repetitions)** - IG - CG	4.73 (0.61, 8.86) 2.53 (−0.05, 5.12)	**0.023** 0.055	5.07 (1.00, 9.12) 1.73 (−1.42, 4.88)	**0.013** 0.258	0.33 (−0.24, 0.91) −0.80 (−4.12, 2.52)	0.409 0.613
**Handgrip strength (kg)** - IG - CG	3.07 (0.62, 5.52) 0.20 (−1.12, 1.52)	**0.013** 1.000	2.41 (0.03, 4.82) 0.54 (−0.75, 1.83)	**0.049** 0.818	−0.65 (−1.92, 0.62) 0.34 (−1.45, 2.13)	0.552 1.000
**Sit-and-reach (right, cm)** - IG - CG	5.42 (−1.13, 11.97) 1.79 (−2.60, 6.17)	0.123 0.397	4.85 (−3.85, 13.55) 0.27 (−4.14, 4.67)	0.455 0.899	−0.57 (−6.16, 5.03) −1.52 (−5.37, 2.33)	1.000 0.411
**Sit-and-reach (left, cm)** - IG - CG	5.45 (−0.07, 10.98) 1.96 (−2.24, 6.16)	0.053 0.334	4.87 (−2.84, 12.58) 1.77 (−2.62, 6.17)	0.197 0.404	−0.59 (−4.51, 3.33) −0.19 (−2.27, 1.88)	0.753 0.845

*Note.*^1^ T1, pre-intervention; ^2^ T2, post-intervention; ^3^ T3, follow-up; ^4^ 95%CI, 95% confidence interval; ^5^ one-way repeated measure ANOVA was performed to detect within-group differences, and post-hoc analysis with Bofferoni correction was employed to test significance of the changes from T1 to T2, T1 to T3, and T2 to T3; ^6^ 6MWT, six-minute walk test; ^7^ IG, intervention group; ^8^ CG, wait-list control group; bold text indicates significant results.

**Table 6 ijerph-19-12015-t006:** Between-group differences in HRPF-related outcomes.

Outcome	IG ^1^ (n = 15) Mean ± SD ^3^	CG ^2^ (n = 15) Mean ± SD	Adjusted Difference ^4^ (95%CI) ^5^	*p*	Partial η^2^
**6MWT ^6^ (m)** - T1 - T2 - T3	417.93 ± 85.12 477.13 ± 76.28 469.67 ± 61.99	453.67 ±105.06 437.60 ± 92.74 444.67 ± 86.14	57.54 (3.20, 111.88) 45.59 (6.63, 84.55)	**0.039** **0.024**	0.149 0.182
**30 s sit-to-stand (repetitions)** - T1 - T2 - T3	12.20 ± 4.21 17.27 ± 5.70 15.67 ± 5.50	13.47 ± 5.33 13.93 ± 6.08 13.40 ± 5.18	4.47 (1.36, 7.57) 3.48 (1.49, 5.47)	**0.006** **0.001**	0.244 0.332
**1-m sit-ups (repetitions)** - T1 - T2 - T3	10.07 ± 8.78 15.60 ± 8.89 14.80 ± 7.74	10.67 ± 11.58 13.20 ± 11.20 12.40 ± 11.98	2.94 (−0.24, 6.11) 2.88 (−1.38, 7.15)	0.068 0.176	0.118 0.069
**Handgrip strength (kg)** - T1 - T2 - T3	15.45 ± 8.87 18.51 ± 6.28 17.86 ± 6.92	16.18 ± 9.31 16.38 ± 9.13 16.72 ± 9.84	2.74 (0.99, 4.49) 1.80 (−0.20, 3.79)	**0.003** 0.076	0.276 0.116
**Sit-and-reach (right, cm)** - T1 - T2 - T3	−0.52 ± 12.49 4.90 ± 6.85 4.33 ± 6.33	−2.76 ± 12.93 −0.97 ± 12.33 −2.49 ± 11.07	4.58 (−0.61, 9.77) 3.70 (−4.15, 11.55)	0.081 0.341	0.108 0.035
**Sit-and-reach (left, cm)** - T1 - T2 - T3	−0.93 ± 12.74 4.52 ± 5.69 3.93 ± 6.45	−4.45 ± 9.27 −2.49 ± 11.18 −2.69 ± 11.71	5.25 (−0.08, 10.58) 5.07 (−1.64, 11.79)	0.053 0.132	0.131 0.085

*Note.*^1^ IG, intervention group; ^2^ CG, wait-list control group; ^3^ SD, standard deviation; ^4^ Adjusted difference, the difference of the intervention group relative to the wait-list control group in ANCOVA, adjusted for covariates (baseline values were adjusted in the between-group difference at T2; baseline values and changes in puberty stage were adjusted in the between-group difference at T3); ^5^ 95%CI, 95% confidence interval; ^6^ 6MWT, six-minute walk test; bold text indicates significant results.

**Table 7 ijerph-19-12015-t007:** Within-group differences in blood pressure.

Outcome	Change from T1 ^1^ to T2 ^2^ (T2–T1)		Change from T1 to T3 ^3^ (T3–T1)		Change from T2 to T3 (T3–T2)	
	Mean Difference (95%CI) ^4^	*p* ^5^	Mean Difference (95%CI)	*p*	Mean Difference (95%CI)	*p*
**Systolic blood pressure (mmHg)** - IG ^6^ - CG ^7^	−6.80 (−14.93, 1.33) −0.53 (−8.98, 7.91)	0.094 0.825	−2.87 (−13.70, 7.96) 1.07 (−11.25, 9.12)	0.579 0.825	3.93 (−6.45, 14.31) −0.53 (−7.12, 6.05)	0.430 0.865
**Diastolic blood pressure (mmHg)** - IG - CG	−2.47 (−15.27, 10.34) −0.93 (−12.85, 10.99)	0.686 0.869	3.53 (−5.06, 12.13) 0.93 (−13.67, 15.54)	0.393 0.893	6.00 (−3.88, 15.88) 1.87 (−8.07, 11.80)	0.214 0.693

*Note.*^1^ T1, pre-intervention; ^2^ T2, post-intervention; ^3^ T3, follow-up; ^4^ 95%CI, 95% confidence interval; ^5^ one-way repeated measure ANOVA was performed to detect within-group differences, and post-hoc analysis with Bofferoni correction was employed to test significance of the changes from T1 to T2, T1 to T3, and T2 to T3; ^6^ IG, intervention group; ^7^ CG, wait-list control group.

**Table 8 ijerph-19-12015-t008:** Between-group differences in blood pressure.

Outcome	IG ^1^ (n = 15) Mean ± SD ^3^	CG ^2^ (n = 15) Mean ± SD	Adjusted Difference ^4^ (95%CI) ^5^	*p*	Partial η^2^
**Systolic blood pressure (mmHg)** - T1 - T2 - T3	127.4 ± 13.61 120.60 ± 10.47 124.53 ± 17.14	127.73 ± 17.88 127.20 ± 14.19 126.67 ± 13.50	−6.48 (−14.91, 1.95) −2.05 (−13.55, 9.45)	0.126 0.716	0.084 0.005
**Diastolic blood pressure (mmHg)** - T1 - T2 - T3	74.73 ± 17.94 72.27 ± 16.68 78.27 ± 12.57	78.8 ± 26.04 77.87 ± 20.22 79.73 ± 16.88	−4.20 (−17.12, 8.72) −0.43 (−11.13, 10.27)	0.511 0.935	0.016 0.001

*Note.*^1^ IG, intervention group; ^2^ CG, wait-list control group; ^3^ SD, standard deviation; ^4^ Adjusted difference, the difference of the intervention group relative to the wait-list control group in ANCOVA, adjusted for covariates (baseline values were adjusted in the between-group difference at T2; baseline values and changes in puberty stage were adjusted in the between-group difference at T3); ^5^ 95%CI, 95% confidence interval.

## Data Availability

The data presented in this study are available upon reasonable request from the first author.

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
