# Peer review of "Effects of a School-Based Physical Activity Intervention for Obesity, Health-Related Physical Fitness, and Blood Pressure in Children with Intellectual Disability: A Randomized Controlled Trial"

_ijerph, 2022, doi:10.3390/ijerph191912015_

Round 1
Reviewer 1 Report
The study description, methodology and results are correct, however the authors could include in the discussion an analysis of the correlation between the modifications of eating habits and the effect of the intervention, as well as the role and influence of the parents in maintaining these behaviors of the young people, specifically for the cognitive condition and the variations of their emotional state.
This is relevant to understand one of the results and it is the little modification in the fat registers of each participant, it would be equally interesting to know the temporal organization of the physical activity sessions, that is to say, they were developed in periods or fixed time frames, they were developed before or after the school activity or during the school activity, also the curriculum of the school of the participants includes some plan of regular physical activity that could have inference in the results. Please include a brief description of the nature of the intellectual disabilities.
Reviewer 2 Report
Thank you for the opportunity to review the manuscript entitled: Effects of a school-based physical activity intervention for obesity, health-related physical fitness, and blood pressure in children with intellectual disability: a randomized controlled trial
This is an interesting topic with a highly quality and relevancy for subjects affected with this condition.
However, I would like to do the following comments:
Line 19. This is the first time in the manuscript using PA, please add the meaning for the general readers.
During the Materials and methods section there are some references listed, however there is not a clear relationship with the methods and some of them for instance, when the authors are describing the dropout rate. Please review and make the changes as needed.
Also, it is not clear the intervention in the waiting-list control. It is not clear if this group received the same treatment, for how long and when the scales were performed. Please clarify
Are there other comorbidities in the group besides Autism and Down Syndrome? Please add a comment.
In line 362, please review the word “coved” seems misspelled.
